# Tegafur–Uracil/Leucovorin Plus Oxaliplatin (TEGAFOX) as Consolidation Regimen after Short-Course Radiotherapy Is Effective for Locally Advanced Rectal Cancer

**DOI:** 10.3390/jcm11102920

**Published:** 2022-05-22

**Authors:** Chun-Kai Liao, Ya-Ting Kuo, Yih-Jong Chern, Yu-Jen Hsu, Yueh-Chen Lin, Yen-Lin Yu, Pao-Shiu Hsieh, Jy-Ming Chiang, Chien-Yuh Yeh, Jeng-Fu You

**Affiliations:** 1Division of Colon and Rectal Surgery, Department of Surgery, Chang Gung Memorial Hospital, Linkou, No. 5, Fuxing Street, Guishan District, Taoyuan 333423, Taiwan; mr9023@cgmh.org.tw (C.-K.L.); kyu52623@cgmh.org.tw (Y.-T.K.); b9202063@cgmh.org.tw (Y.-J.C.); m8295@cgmh.org.tw (Y.-J.H.); ld@cgmh.org.tw (Y.-C.L.); hsiehps@yahoo.com (P.-S.H.); jmjiang@adm.cgmh.org.tw (J.-M.C.); chnyuh@gmail.com (C.-Y.Y.); 2Division of Colon and Rectal Surgery, Department of Surgery, Chang Gung Memorial Hospital, Keelung Branch, No. 222, Maijin Road, Anle District, Keelung City 204201, Taiwan; tomyuauk@cgmh.org.tw; 3School of Medicine, Chang Gung University, No. 259, Wenhua 1st Road, Guishan District, Taoyuan 333323, Taiwan

**Keywords:** short-course radiotherapy, locally advanced rectal cancer, consolidation chemotherapy, neoadjuvant rectal score, TEGAFOX, FOLFOX

## Abstract

This study aimed to explore the safety and efficacy of neoadjuvant SCRT and tegafur–uracil/leucovorin plus oxaliplatin (TEGAFOX) for LARC in comparison to those of the modified 5-fluorouracil, leucovorin, and oxaliplatin (mFOLFOX-6) regimen. We retrospectively evaluated 15 and 22 patients with LARC who underwent SCRT, followed by consolidation chemotherapy with TEGAFOX and mFOLFOX-6 before surgery, respectively, between January 2015 and December 2019. The primary endpoint was the tumor response rate. The secondary endpoints were compliance, toxicity, complications, overall survival (OS), and disease-free survival (DFS). The dose reduction rate was lower in the TEGAFOX group (0 vs. 9.1% (*n* = 2)). No grade III-IV toxicities occurred in the TEGAFOX group. Two and four patients in the TEGAFOX and mFOLFOX-6 groups, respectively, achieved clinical complete responses. The pathologic complete response rate was lower in the TEGAFOX group (7.7% vs. 17.6%). Overall, 11 (73.3%) and 17 (81.0%) patients had a neoadjuvant rectal (NAR) score of <16 in the TEGAFOX and mFOLFOX-6 groups, respectively. All patients in this study received sphincter-preservation surgery. One patient in each group developed Clavien–Dindo grade III complications. There were no significant between-group differences in the 3-year OS (81.8% vs. 84.8%, *p* = 0.884) and 3-year DFS (72% vs. 71.6%, *p* = 0.824) rates. TEGAFOX, as consolidation chemotherapy after SCRT, achieves good tumor downstaging and patient compliance in LARC. The toxicity, complications, and surgical outcomes are similar to those of mFOLFOX-6. Thus, TEGAFOX can be considered a chemotherapy option for rectal cancer treatment.

## 1. Introduction

Colorectal cancer (CRC) is the third leading cause of cancer and the second leading cause of cancer-related deaths worldwide. The GLOBOCAN statistics showed that more than 1.9 million incident CRC cases were recorded in 2020, with 39% of the tumors located in the rectum [1]. Neoadjuvant (chemo)radiation followed by total mesorectal excision (TME) is currently the standard management modality for locally advanced rectal cancer (LARC) [2,3]. Two major neoadjuvant radiotherapy techniques are currently widely used: short-course radiotherapy with immediate surgery in 1 week or long-course radiotherapy with concurrent chemotherapy followed by surgery after an interval of 4–6 weeks [4,5]. Previous studies have reported similar local control, toxicity, and survival between these two approaches [6,7]. However, despite good local control, distal metastasis is still reported in up to 30% of the patients [7,8], and thus, a new treatment strategy to reduce early metastasis is urgently needed.

Total neoadjuvant therapy (TNT) was recently proposed for the treatment of LARC with the application of systemic chemotherapy in the neoadjuvant setting. The concept of TNT includes the early use of chemotherapy for the eradication of micrometastasis and the application of treatment in patients with better compliance. In the TNT approach, either short-course radiotherapy or long-course chemoradiation is used in combination with systemic chemotherapy [9]. Considering convenience, better patient compliance, and lower medical costs, short-course radiotherapy is favored in many northern European countries. The Polish II study first reported the use of neoadjuvant SCRT followed by three cycles of FOLFOX-4 chemotherapy before surgery for LARC in 2016. Importantly, it showed an equivalent local efficacy to conventional chemoradiotherapy (CRT) and a better 3-year overall survival (OS) [10]. The RAPIDO trial recently reported that compared with conventional CRT, SCRT followed by either six cycles of CAPOX or nine cycles of FOLFOX-4 before TME surgery achieves better pathological complete response (pCR) and lower disease-related treatment failure (DrTF) [11]. The addition of oxaliplatin to neoadjuvant chemotherapy has been established to improve pathological response [12,13], and several 5-fluorouracil (5-FU) derivatives are available to be combined with oxaliplatin, including infusion 5-FU (FOLFOX regimen) or capecitabine (CAPOX or XELOX regimen).

Tegafur-uracil (UFT)/leucovorin is another 5-FU derivative that is commonly used as an adjuvant treatment after curative surgery for stage II/III colon cancer or metastatic colorectal cancer (mCRC) [14,15]. The TEGAFOX regimen, which is composed of UFT/LV and oxaliplatin, has been proven to be effective for both unresectable mCRC and adjuvant treatment [16,17]. However, data regarding the use of this regimen in the neoadjuvant setting for rectal cancer are limited. Thus, this study aimed to explore the safety and efficacy of TEGAFOX after SCRT as a neoadjuvant treatment for LARC in comparison with those of the modified 5-fluorouracil, leucovorin, and oxaliplatin (mFOLFOX-6) regimen. The primary endpoint was the tumor response rate. The secondary endpoints were compliance, toxicity, complications, overall survival (OS), and disease-free survival (DFS).

## 2. Material and Methods

### 2.1. Study Design and Patients

This was a retrospective study of patients with locally advanced rectal cancer who underwent SCRT followed by consolidation chemotherapy before surgery at the Colorectal Section Tumor Registry of Chang Gung Memorial Hospital between January 2015 and December 2019. The inclusion criteria were as follows: (1) pathologically confirmed rectal adenocarcinoma; (2) the distal edge of the tumor located ≤10 cm from the anal verge; (3) clinical stage with T3-4 or N+ tumor; (4) underwent at least two cycles of consolidation chemotherapy between completion of radiotherapy and surgery; (5) no synchronous distant metastasis before neoadjuvant treatment; and (6) the consolidation chemotherapy regimen was TEGAFOX or FOLFOX. Data were collected from the prospectively designed database consisting of records of postoperative patients who were consecutively and actively followed up. The patient selection flowchart is shown in Figure 1.

This retrospective chart review study involving human participants was in accordance with the ethical standards of the institutional and national research committee and with the 1964 Helsinki Declaration and its later amendments or comparable ethical standards. The Human Investigation Committee (IRB) of Chang Gung Memorial Hospital approved this study (approval number 202101444B0). All data were recorded in the hospital database and were used for research purposes only. Informed consent was waived owing to the retrospective nature of the study.

### 2.2. Treatment Protocols

The treatment protocol for all patients was discussed at the multidisciplinary team meeting (MDT).

### 2.3. Radiotherapy

SCRT comprised 5 consecutive fractions of 5 Gy delivered using three-dimensional conformal radiotherapy or volumetric modulated arc therapy with 6 or 10 MV photon X-rays. The clinical target volume included the tumor, which involved regional lymph nodes, elective pelvic lymph nodes, and the entire mesorectum with adequate margins.

### 2.4. Chemotherapy

Consolidation chemotherapy was commonly initiated 1 week after the completion of SCRT. The TEGAFOX regimen comprised oxaliplatin 85 mg/m^2^ on day 1, tegafur-uracil 300 mg/m^2^/day, and leucovorin 90 mg/day, administered on days 1 to 5 and 8 to 12. Each cycle was repeated every 2 weeks for 4 to 6 cycles. The mFOLFOX-6 regimen comprised oxaliplatin 85 mg/m^2^ on day 1, leucovorin 400 mg/m^2^ on day 1, bolus 5-FU 400 mg/m^2^ on day 1 and then a 5-FU infusion (2400 mg/m^2^) for 46 hr. Each cycle was repeated every 2 weeks for 4 to 6 cycles. The consolidation chemotherapy regimen was determined at the discretion of the primary care physician after discussion with the patients about the above chemotherapy regimen, the routes of drug administration, and the frequency of hospital visits (Figure 2). Before starting the consolidation chemotherapy, the physicians usually offer two kinds of regimens, TEGAFOX, or FOLFOX, and discuss with the patients to determine which regimen is to be used. Suppose the patient receives a continuous IVF infusion, either during the hospitalization (which needs a 3-day hospital stay every two weeks) or takes the infusor pump back home (which needs two hospital visits in 3 days every two weeks). In this case, the FOLFOX regimen will be chosen. On the other hand, if the patient wishes to have fewer hospital stays and visits, the TEGAFOX regimen will be suggested (only one hospital visit every two weeks is required).

After surgery, an MDT reviewed the final pathological stage of the patient and decided on the need for adjuvant chemotherapy. The common adjuvant chemotherapy regimen was mFOLFOX-6 for a total of 12 cycles (including consolidation chemotherapy) and started 4 to 6 weeks after the surgery.

### 2.5. Surgery

After the completion of neoadjuvant treatment, the tumor stage was re-evaluated by colonoscopy and imaging survey. Extirpative surgery mainly included a lower anterior resection (LAR), Hartmann’s procedure, or abdominal perineal resection, determined according to the re-stage of the tumor and performed after discussion at the MDT meeting. The choice between laparoscopy and laparotomy was determined based on the surgeon’s preference. A protective colostomy/ileostomy was created based on the surgeon’s judgment.

### 2.6. Assessment

All patients underwent complete staging before treatment initiation. The staging workup included physical examination, digital rectal examination, and colonoscopy with or without rectal endoscopic ultrasonography. A chest, abdominal, and pelvic computed tomography (CT) survey was used to assess the tumor and rule out distant organ metastasis. Pelvic magnetic resonance imaging (MRI) was performed to evaluate regional tumor conditions. Re-staging after neoadjuvant treatment usually involved physical examination, colonoscopy, and CT or MRI. The staging was conducted according to the 8th edition of the Union for International Cancer Control TNM Classification. After surgery, the tumor regression grade (TRG) of the specimen was defined according to the Rödel grading system [18]. Clinical complete response (cCR) was defined as no tumor at the primary lesion site on imaging and colonoscopy re-evaluation and no recurrence/metastasis within at least 1 year of follow-up. The neoadjuvant rectal (NAR) score was also calculated to assess the tumor response as follows: NAR = [5pN – 3(cT – pT) + 12]^2^/9.61 [19]. The tumor downstaging was categorized as increased, decreased, or no change according to the difference between the clinical stage and pathological stage.

For morbidity and toxicity evaluation, the Common Terminology Criteria for Adverse Events, version 5, was used to evaluate toxicity during neoadjuvant radiotherapy and chemotherapy. Postoperative complications were defined as any morbidity within 30 days after surgery. The Clavien–Dindo classification was used to grade surgical complications [20].

### 2.7. Statistical Analysis

Categorical variables were compared using Pearson’s chi-squared test or Fisher’s exact test, while continuous variables were compared using the Mann–Whitey U test. Survival curves were generated using the Kaplan–Meier method and compared between the two groups using the log-rank test. The index date of survival analysis was the day the patient started radiotherapy. All statistical analyses were performed using the Statistical Package for Social Sciences (SPSS), version 24 (IBM Corp., New York, NY, USA) and GraphPad Prism version 9 (GraphPad Software Inc., San Diego, CA, USA). Differences were considered statistically significant at a two-sided *p*-value of < 0.05.

## 3. Results

### 3.1. Patient Characteristics

A total of 37 consecutive patients with rectal cancer who underwent SCRT and consolidation chemotherapy were enrolled in this study. The mean age was 56.27 ± 9.78 years and 61.82 ± 10.01 years in TEGAFOX and mFOLFOX-6 groups, respectively. Lower rectal tumors were seen in 80% and 63.6% of patients in the TEGAFOX and mFOLFOX-6 groups, respectively. There were no significant between-group differences in the clinical T-stage, N-stage, pre-treatment carcinoembryonic antigen (CEA), post-treatment CEA, and tumor histology grade. The patient characteristics are summarized in Table 1.

### 3.2. Pathology and Tumor Response 

The pathological findings are summarized in Table 2. Two patients in the TEGAFOX group and four patients in the mFOLFOX-6 group did not undergo surgery due to a clinical complete response. One patient in the mFOLFOX-6 group had no surgical management due to distant metastasis occurring during neoadjuvant treatment and received further chemotherapy. A total of 30 patients underwent surgery for the rectal tumor. Of them, pCR was observed in 7.7% and 17.6% of the patients in the TEGAFOX and mFOLFOX-6 groups, respectively. Good response (TRG 3–4) was observed in 61.5% and 64.7% of the patients in the TEGAFOX and mFOLFOX-6 groups. There was no significant between-group difference in the circumferential margin, R0 resection rate, or tumor downstaging. Two and four patients in the TEGAFOX and mFOLFOX-6 groups, respectively, achieved cCR. The mean NAR score was 16.50 ± 14.81 in the TEGAFOX group and 15.26 ± 18.40 in the mFOLFOX-6 group. There were 11 (73.3%) and 17 (81.0%) patients who had NAR scores less than 16 in each group.

### 3.3. Acute Toxicity and Compliance

All patients in both groups completed the full dose of radiotherapy without any dose reduction or delay. The median interval between the initiation of radiotherapy and surgery was 10.71 weeks and 16.28 weeks in the TEGAFOX and mFOLFOX-6 groups, respectively. No patient in the TEGAFOX group required a dose reduction of chemotherapy, whereas two (9.1%) patients in the mFOLFOX-6 group required dose reductions. Chemotherapy was postponed due to grade 3 neutropenia and due to vomiting and dizziness in one patient each in the mFOLFOX-6 group. The most common grade 1–2 toxicity in both groups was diarrhea, followed by neuropathy. No grade 4–5 toxicity was observed in the study cohort. There was a difference in chemotherapy cycles: 46.6% of the patients in the TEGAFOX group received four cycles, while 72.7% of the patients in the mFOLFOX-6 group received six cycles (chemotherapy <12 weeks: 60% vs. 18.2%, *p* = 0.015). Overall, 86.7% and 95.5% of the patients in the TEGAFOX and mFOLFOX-6 groups, respectively, received at least 8 weeks of consolidation chemotherapy. The details of toxicity and compliance are summarized in Table 3.

### 3.4. Surgery and Postoperative Complications

The details of the surgery and postoperative complications are listed in Table 4. All patients in the study cohort underwent sphincter-preservation surgery. Two and one patients in the TEGAFOX and mFOLFOX-6 groups, respectively, underwent local excision after neoadjuvant treatment. Overall, 38% of the patients in the TEGAFOX group underwent LAR with intersphincteric resection, while 47.1% of the patients in the mFOLFOX-6 group underwent LAR with trans-anal total mesorectal excision. In addition, 69% and 84.2% of the patients in the TEGAFOX and mFOLFOX-6 groups, respectively, underwent laparoscopic surgery. A protective stoma was performed in all patients in the TEGAFOX group and in 68.8% of the patients in the mFOLFOX-6 group. There was one grade 3 postoperative complication in each group: the patients developed anastomosis leakage and underwent surgery for diverting stoma. In both groups, the most common complication was urine retention, and all the patients received conservative treatment. No patient died postoperatively.

### 3.5. Follow-Up and Medium-Term Survival

The median follow-up duration was 34.6 (range, 13.6 to 55.2) and 32.5 (range, 7.4 to 65.4) months in the TEGAFOX and mFOLFOX-6 groups, respectively. One patient in the TEGAFOX group and two patients in the mFOLFOX-6 group had a local recurrence. There was no significant between-group difference in distant metastasis (26.7% vs. 30%). No distant metastasis or local recurrence was observed in the six patients with cCR during the study period. The 3-year OS (81.8% vs. 84.8%, *p* = 0.884) and 3-year disease-free survival (DFS) (72% vs. 71.6%, *p* = 0.824) were not significantly different between the TEGAFOX and mFOLFOX-6 groups (Figure 3).

## 4. Discussion

Neoadjuvant therapy is a crucial treatment modality for LARCs. In most countries, conventional long-course chemoradiotherapy is recommended for LARC, despite several studies showing that SCRT achieves similar efficacy [6,7]. The main issue with using SCRT in LARC is the shorter interval between RT and surgery, limiting the tumor response and preventing sphincter-preservation surgery. With respect to the tumor response from a longer interval after SCRT, the Stockholm III trial revealed a higher rate if surgery was performed after 4–8 weeks than if it was performed immediately (10.1% vs. 1.7%) [21]. However, improved local control did not lead to a better long-term prognosis. OS and DFS did not differ within 2 years of follow-up [22]. Although a higher pCR rate was observed if surgery was delayed after SCRT, a new concern is that early distant metastasis could occur during the prolonged interval of surgery. Thus, conventional CRT, which is composed of concurrent 5-FU-based chemotherapy, is still preferred by many doctors because it offers the benefit of chemotherapy before surgery. Nevertheless, the distant metastasis rate is still as high as 30% in traditional neoadjuvant treatment (NAT) [7,8,23]. As such, the addition of systemic chemotherapy in neoadjuvant treatment, that is, total neoadjuvant treatment (TNT), was then proposed for the purpose of early eradication of micrometastasis.

There is currently no consensus on the use of SCRT or conventional CRT for TNTs. The phase II Dutch trial first reported that SCRT followed by systemic chemotherapy can achieve a 72% R0 resection rate and 4.4 years of median OS in patients with stage IV rectal cancer [24,25]. In recent decades, several studies have revealed that SCRT followed by systemic oxaliplatin-based chemotherapy achieves a better downstaging effect and comparable prognosis in patients with LARC [10,11,26,27]. In a meta-analysis by Patel et al., SCRT achieved a higher pCR rate than did long-course CRT (RR: 1.75, 95% CI: 1.41–2.19) [28]. Further, SCRT achieved comparable OS, R0 resection, and T-downstaging; however, acute toxicity was also higher in the SCRT + chemotherapy than in CRT alone.

In Taiwan, the National Health Insurance (NHI) reimbursement only covers the 5-FU infusion or tegafur-uracil for stage II-III rectal cancer [29]. The patients need to pay USD 200 for each cycle of oxaliplatin administration, and another USD 200 is required for each cycle of capecitabine (10 days in 2 weeks as one cycle). As such, doctors tend to use a single agent as a radiosensitizer in combination with conventional CRT for LARC treatment. With the growing evidence showing better local efficacy of SCRT followed by consolidation chemotherapy in LARC, doctors in our institution gradually adjusted the NAT strategy into this new approach. There are many benefits of using SCRT over long-course RT, including patient convenience owing to fewer hospital visits, lower medical costs, and even better cost-effectiveness in the long-term follow-up [30]. To minimize the need for hospital visits, using an oral 5-FU derivative with 1 day of oxaliplatin infusion is a good strategy.

To the best of our knowledge, this is the first study to compare TEGAFOX with mFOLFOX-6 as a neoadjuvant treatment for LARC. Regarding radiotherapy compliance, no patients had reduced dose or delayed SCRT, consistent with previous reports that SCRT can reach a 100% completion rate [10,31]. For compliance with chemotherapy, only one patient who developed grade 3 neutropenia required a dose reduction in the mFOLFOX-6 group, and no grade 3 adverse events (AEs) occurred in the TEGAFOX group, indicating the safety of the regimen. Several studies showed that compared with 5-FU alone in conventional CRT, the addition of oxaliplatin to consolidation chemotherapy is associated with a higher rate of acute toxicity. The acute toxicity rate varies across studies. Markovina et al. reported a significantly higher rate of grade 3–4 hematologic toxicity in patients with four cycles of mFOLFOX-6 (22% vs. 0%, *p* < 0.001) [31]. In the recent RAPIDO trial, grade 3–4 AEs were more common in the treatment arm than in the control arm (48% vs. 22%) [32]. In contrast, Aghili et al. reported no difference in grade ≥3 AEs (24.2% vs. 22.2%, *p* = 0.55) [33]. Chakrabarti et al. also reported comparable grade ≥ 3 GI toxicities (2% vs. 4%, *p* = 1) [34]. Both studies evaluated 3–4 weeks of XELOX as consolidation chemotherapy. AEs may be related to the regimen and cycles of chemotherapy.

Regarding tumor response, six patients (16.2%) achieved cCR after NAT and for at least 1 year. Among the patients who underwent curative resection surgery, four patients (13.3%) achieved pCR, resulting in a total response rate (cCR + pCR) of 27%. This result is comparable with current pCR rates of 7.1%–35% in SCRT followed by consolidation chemotherapy before TME surgery [28]. Overall, good tumor and lymph node downstaging were observed in this study. When examining the pCR rate according to different chemotherapy regimens, a relatively lower rate was observed in the TEGAFOX group. A possible reason is that shorter cycles of TEGAFOX were used in the study cohort, and thus, there was a shorter interval between RT and surgery. This limited the time span for tumor response. However, the mean NAR score and distribution in the two groups were not different.

All the patients in this study underwent sphincter-preservation surgery and had a low rate of postoperative complications. Only two patients in each group required reoperation for diverting stoma due to leakage. This result is consistent with previous reports that SCRT and consolidation chemotherapy did not increase the rate of postoperative complications [31,32,33,34]. Particularly, 81% of the patients in our study had tumors located <5 cm from the anal verge, and our results revealed a feasible approach for downstaging and organ preservation for lower rectal tumors. Within a median follow-up time of 34 months, the 3-year OS and DFS rates were 81.8% and 72.0%, respectively, in the TEGAFOX group and were 84.8% and 71.6%, respectively, in the mFOLFOX-6 group. The medium-term survival in the present study is similar to that reported in other larger trials. The Polish II study reported a 3-year OS rate of 73% and a 3-year DFS rate of 53% [10]. In addition, the 3-year OS and DFS rates in the RAPIDO trial were 89.1% and 76%, respectively [11]. In the recent STELLAR trial, the 3-year OS was 86.5% and the 3-years DFS was 64.5% [35]. Diverse study protocols may result in different survival rates, but the results are consistent in that the prognostic benefit of SCRT is non-inferior to that of conventional CRT.

Although good local efficacy was observed, distant metastasis remained the main cause of a decreased DFS in the current study. In total, 26.7% and 30% of the patients in the TEGAFOX and mFOLFOX-6 groups, respectively, experienced at least one distant organ metastasis. The RAPIDO study recently showed better DrTF in the experimental group, in which patients received six cycles of CAPOX or nine cycles of FOLFOX-4, than in the conventional CRT group. However, the majority of the patients who had DrTF had distant metastases. The cumulative probability of distant metastases was 20.0% at the 3-year follow-up [11]. In the Polish II study, which involved shorter cycles of consolidation chemotherapy, the 3-year cumulative incidence of distant metastases was 30% [10]. Collectively, these findings indicate that the prognosis of LARC is not influenced by the consolidation chemotherapy regimen. Although treatment compliance is better in consolidation chemotherapy than in adjuvant treatment, the rate of distant metastasis remains high in TNT, for unclear reasons. Further studies are needed to explore these findings.

Tegafur-uracil is a 5-FU derivative that has been proven to have comparable efficacy to 5-FU in either mCRC or high-risk stage II/III colon cancer [14,15]. Bennouna et al. administered the TEGAFOX regimen (tegafur-uracil, leucovorin, and oxaliplatin) as first-line treatment for mCRC and reported a 15% rate of grade 3–4 peripheral neuropathy and a 10% rate of grade 3–4 neutropenia without any complications. The authors concluded that this regimen is safe and can be an alternative to intravenous infusion [16]. At our institution, we typically use UFT as the radiosensitizer in the conventional CRT setting; oxaliplatin plus infusion 5-FU, capecitabine, or UFT is selected as adjuvant chemotherapy for high-risk stage II/III CRC or mCRC. With increasing evidence that adding oxaliplatin to NAT can induce a better tumor response, some doctors have started to introduce oxaliplatin into the pre-existing treatment strategy for LARC. Compared with the mFOLFOX-6 regimen, the TEGAFOX regimen is more convenient for patients because it minimizes the frequency of hospital visits, thus markedly reducing transportation costs. Regarding the cycles of consolidation chemotherapy, a wide range of between 2 to 9 cycles was reported worldwide. All studies showed a good tumor response and had comparable oncological outcomes. At my institution, the doctors first choose four cycles of TEGAFOX as a consolidation chemotherapy regimen in LARC. With growing evidence showing the safety and efficacy of tumor response after using six or more cycles of the FOLFOX regimen, prolonged chemotherapy cycles, mostly six cycles, was adopted gradually. A recent phase II randomized controlled trial by Kosugi et al. showed that TEGAFOX is a feasible adjuvant chemotherapy regimen for high-risk stage II/III CRC, achieving better DFS with manageable AEs compared to oral UFT alone [17]. The present study shows that neoadjuvant SCRT followed by consolidation TEGAFOX chemotherapy resulted in good tumor response, patient compliance, and controllable side effects and complications for patients with LARC. These findings provide evidence that TEGAFOX is a feasible regimen for neoadjuvant chemotherapy for rectal cancer.

To our best knowledge, this study is the first to compare TEGAFOX with FOLFOX as a neoadjuvant treatment for LARC. However, there are some limitations that need to be considered. First, this was a retrospective study, and thus, the possibility of selection bias could not be ruled out. Second, the sample size was small, and the follow-up time was inadequate to evaluate long-term prognosis and toxicity. In addition, the differences between the TEGAFOX and FOLFOX cycles may have influenced the results. Further prospective studies are required to confirm our findings.

## 5. Conclusions

TEGAFOX consolidation chemotherapy after SCRT in the neoadjuvant setting is safe and effective for tumor downstaging in LARC. The associated toxicity, complications, and surgical outcomes are comparable to those of the mFOLFOX-6 regimens. Thus, TEGAFOX can be considered a chemotherapy option for rectal cancer treatment.

## Figures and Tables

**Figure 1 jcm-11-02920-f001:**
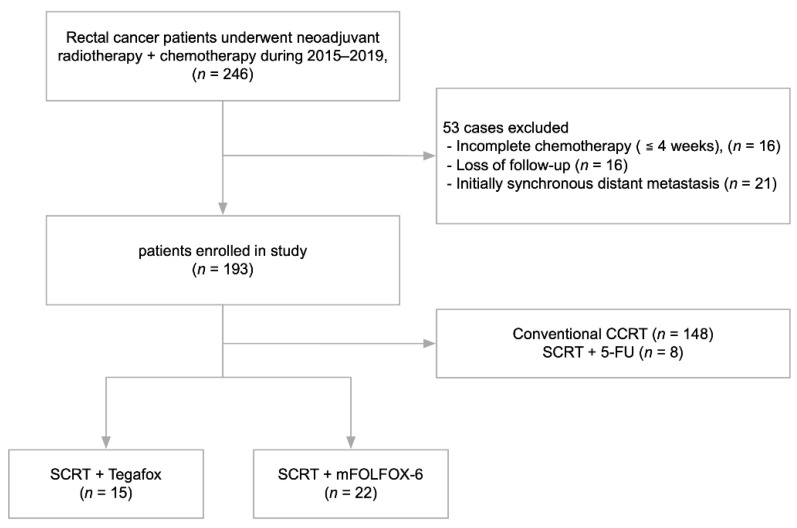
Patient selection flowchart. CCRT: concurrent chemoradiotherapy; SCRT: short-course radiotherapy.

**Figure 2 jcm-11-02920-f002:**
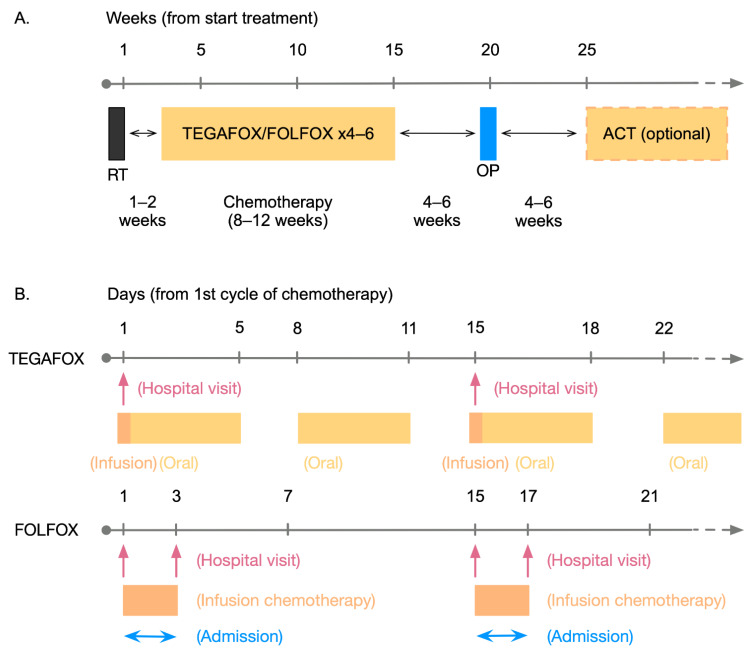
(**A**) The timeline of each individual therapeutic intervention in this study; (**B**) The detailed timeline of the 1st and 2nd cycle of consolidation chemotherapy. RT: radiotherapy; OP: operation; ACT: adjuvant chemotherapy.

**Figure 3 jcm-11-02920-f003:**
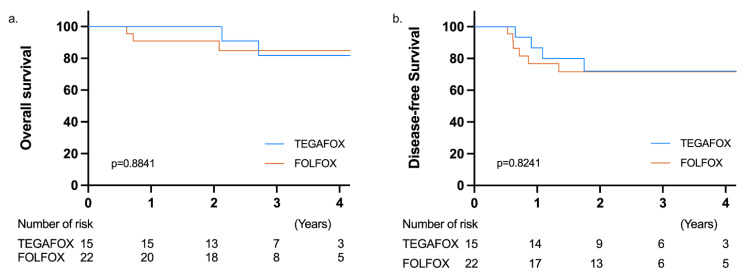
Comparison of Kaplan–Meier survival curves between the TEGAFOX and mFOLFOX-6 groups. (**a**) Overall survival (OS). (**b**) Disease-free survival (DFS).

**Table 1 jcm-11-02920-t001:** Patient characteristics by chemotherapy regimen.

	TEGAFOX (*n* = 15)	mFOLFOX-6 (*n* = 22)	*p* Value
Age at diagnosis (years)	56.27 ± 9.78	61.82 ± 10.01	0.209
Sex			
Female	3 (20)	9 (40.9)	0.286
Male	12 (80)	13 (59.1)	
ECOG-PS score *			
0	1 (6.7)	4 (18.2)	0.815
1	13 (86.7)	17 (77.3)	
2	1 (6.7)	1 (4.5)	
Clinical T stage			
T2	3 (20)	5 (22.7)	1
T3	9 (60)	12 (54.5)	
T4	3 (20)	5 (22.7)	
Clinical N stage			
N0	5 (33.3)	7 (31.8)	1
N+	10 (66.7)	15 (68.2)	
Tumor distance from the AV (cm)			
≤5	12 (80)	14 (63.6)	0.285
>5	3 (20)	8 (36.4)	
Pretreatment CEA (ng/mL)			
<5.0	7 (46.7)	15 (68.2)	0.191
≥5.0	8 (53.3)	7 (31.8)	
Posttreatment CEA (ng/mL)			
<5.0	13 (92.9)	21 (95.5)	1
≥5.0	1 (7.1)	1 (4.5)	
Clinical stage			
I	2 (13.3)	3 (13.6)	1
II	3 (20)	4 (18.2)	
III	10 (66.7)	15 (68.2)	
Histology grade **			
Well	3 (20)	4 (18.2)	0.863
Moderate	10 (66.7)	13 (59.1)	
Poor	2 (13.3)	1 (4.5)	

* ECOG-PS: Eastern Cooperative Oncology Group Performance Status; ** some data lost.

**Table 2 jcm-11-02920-t002:** Pathology findings by chemotherapy regimen.

	TEGAFOX (*n* = 13) *	mFOLFOX-6 (*n* = 17) *	*p* Value
T stage			
ypT0	1 (7.7)	3 (17.6)	0.944
ypT1	1 (7.7)	1 (5.9)	
ypT2	4 (30.8)	4 (23.5)	
ypT3	6 (46.2)	7 (41.2)	
ypT4	1 (7.7)	2 (11.8)	
N stage			
ypN0	11 (84.6)	13 (76.5)	0.803
ypN1	1 (7.7)	1 (5.9)	
ypN2	1 (7.7)	3 (17.6)	
CRM			
Positive	0	4 (23.5)	0.113
Negative	13	13 (76.5)	
Stage **			
ypT0N0	1 (7.7)	3 (17.6)	0.814
yp stage I	5 (38.5)	4 (23.5)	
yp stage II	5 (38.5)	5 (29.4)	
yp stage III	2 (15.4)	4 (23.5)	
TRG			
Grade 0	1 (7.7)	1 (5.9)	0.872
Grade 1	2 (15.4)	1 (5.9)	
Grade 2	2 (15.4)	4 (23.5)	
Grade 3	7 (53.8)	8 (47.1)	
Grade 4	1 (7.7)	3 (17.6)	
Perineural invasion			
Positive	2 (15.4)	2 (11.8)	1
Negative	11 (84.6)	15 (88.2)	
Lymphovascular invasion			
Positive	1 (7.7)	4 (23.5)	0.355
Negative	12 (92.3)	13 (76.5)	
R0 resection	11 (84.6)	13 (76.5)	0.672
NAR score	16.50 ± 14.81	15.26 ± 18.40	0.474
<16	11 (73.3)	17 (81.0)	0.694
≥16	4 (26.7)	4 (19.0)	
cCR	2 of 15	4 of 22	
pCR + cCR	3 (20)	7 (31.8)	
T-Downstaging			
decreased	6 (46.2)	8 (47.1)	0.760
unchanged	5 (38.5)	8 (47.1)	
increased	2 (15.4)	1 (5.9)	
N-Downstaging			
decreased	6 (46.2)	8 (47.1)	0.695
unchanged	6 (46.2)	9 (52.9)	
increased	1 (7.7)	0	

* Two patients in the TEGAFOX group and four patients in the mFOLFOX-6 group had no surgery due to clinical complete response. One patient in the mFOLFOX-6 group had no surgery due to distant metastasis found during neoadjuvant treatment and received further chemotherapy. ** One patient in the mFOLFOX-6 group had distant metastasis during neoadjuvant treatment.

**Table 3 jcm-11-02920-t003:** Patient compliance and acute toxicity by chemotherapy regimen.

	TEGAFOX (*n* = 15)	mFOLFOX-6 (*n* = 22)	*p* Value
Interval between RT and CT completion (weeks), median	7.57	11.21	0.026
Interval between CT completion and surgery (weeks), median	4.43	4.71	0.378
Interval between RT and surgery (weeks), Median *	10.71	16.28	0.065
Completion of full-dose radiotherapy	15 (100)	22 (100)	1
Delay/dose reduction of radiotherapy	0 (0)	0 (0)	1
Duration of consolidation chemotherapy (weeks)			
<12	9 (60.0)	4 (18.2)	0.015
≥12	6 (40.0)	18 (81.8)	
Completion of full-dose chemotherapy (8 weeks)	13 (86.7)	21 (95.5)	0.554
Delay/dose reduction of chemotherapy	0	2 (9.1)	0.505
Grade 1–2 toxicity			
Neurology	2	5	
Neutropenia	0	1	
Diarrhea	4	6	
Fatigue	0	3	
Skin	0	2	
Grade 3–4 toxicity			
Neutropenia	0	1	

* Seven patients who did not undergo surgery are excluded.

**Table 4 jcm-11-02920-t004:** Operative characteristics and postoperative complications by chemotherapy group.

	TEGAFOX (*n* = 13)	mFOLFOX-6 (*n* = 17)	*p* Value
Type of operation			
LAR	6 (46.2)	8 (47.1)	0.01
LAR + ISR	4 (30.8)	0	
LAR + TaTME	1 (7.7)	8 (47.1)	
Local excision	2 (15.4)	1 (5.9)	
Surgical approach			
Laparotomy	2 (15.4)	2 (11.8)	0.693
Laparoscopy	9 (69.2)	14 (84.2)	
Trans-anal only	2 (15.4)	1 (5.9)	
Protective stoma **	11 (100)	11 (68.8)	0.06
Blood loss (mL)	105 ± 118	207 ± 258	0.314
Operative time (min)	337 ± 176	404 ± 174	0.250
Postoperative complications			
Anastomosis leakage	1	1	
Urine retention	0	3	
Ileus	0	1	
SSI	0	1	
Clavien-Dindo grade			
I–II	0	4	0.5
III–IV	1	1	

LAR, low anterior resection; ISR, intersphincteric resection; TaTME, trans-anal total mesorectal excision; SSI, surgical site infection; ** Excluding 3 patients with local excision only.

## Data Availability

The datasets generated and analyzed during the current study are available from the corresponding author upon reasonable request.

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
