# Peer review of "Tegafur–Uracil/Leucovorin Plus Oxaliplatin (TEGAFOX) as Consolidation Regimen after Short-Course Radiotherapy Is Effective for Locally Advanced Rectal Cancer"

_jcm, 2022, doi:10.3390/jcm11102920_

Round 1
Reviewer 1 Report
Excellently written paper, minor comments:
Pls include abbreviations in flow chart caption (fig 1)
P3 L115f "The consolidation chemotherapy regimen was determined at the
discretion of the primary care physician." which criteria?
Interval btw RT and CT appears very different for TEGAFOX and mFOLFOX, though not significant (table 2): pls discuss further
Author Response
We deeply appreciate these kindly and creative feedback and comment from distinguished reviewer. Below are our response point-by-point.
Excellently written paper, minor comments:
- Pls include abbreviations in flow chart caption (fig 1)
Response: I had added the abbreviations in the figure caption.
- P3 L115f "The consolidation chemotherapy regimen was determined at the
discretion of the primary care physician." which criteria?
Response: Before starting the consolidation chemotherapy, the physicians usually offer two kinds of regimens, TEGAFOX or FOLFOX, and discuss with the patients to determine which regimen to be used. Suppose the patient can receive a continuous IVF infusion, either during the hospitalization (which needs a 3-day hospital stay every two weeks) or bring the infusor pump back home (which needs two hospital visits in 3 days every two weeks). In that case, the FOLFOX regimen will be chosen. On the other hand, if the patient wishes to have fewer hospital stays and visits, the TEGAFOX regimen will be suggested (only one hospital visit every two weeks is required). Thanks for the reviewer’s kindly feedback, and I have revised the manuscript in the methods paragraph adding the below explanation: “The consolidation chemotherapy regimen was determined at the discretion of the primary care physician after discussion with the patients about the above chemotherapy regimen, the routes of drug administration, and the frequency of hospital visits”.
- Interval btw RT and CT appears very different for TEGAFOX and mFOLFOX, though not significant (table 2): pls discuss further
Response: As another reviewer’s suggestion, I had revised the manuscript and re-evaluated all the continuous variables with Mann–Whitney U test. After using the non-parametric test, we found a significant difference between the RT and CT intervals for TAGAFOX and mFOLFOX-6. To explain this difference, I revised the manuscript in the discussion section below: “Regarding the cycles of consolidation chemotherapy, a wide range between 2 to 9 cycles was reported worldwide. All studies showed a good tumor response and had comparable oncological outcomes. At my institution, the doctors first choose four cycles of TEGAFOX as a consolidation chemotherapy regimen in LARC. With growing evidence showing the safety and efficacy of tumor response after using 6 or more cycles of the FOLFOX regimen, prolonged chemotherapy cycles, mostly 6 cycles, was adopted gradually.”
Moreover, in the original manuscript, the limitation section also mentioned that the differences between the TEGAFOX and FOLFOX cycles may have influenced the results.
Reviewer 2 Report
Chun-Kai Liao et al. examined the feasibility of tegafur–uracil/leucovorin plus oxaliplatin (TEGAFOX) as a consolidation regimen after short-course radiotherapy for locally advanced rectal cancer (LARC). The authors reported acceptable safety profile and accouraging oncological outcomes. However, some of the aspects listed below require attention.
- The abstract defined tumor downstaging as the primary endpoint, however this aspect is omitted in the remaining part of the manuscript, particularly methods and results.
- No. 4 of inclusion criteria (at least 4 weeks of interval between completion of radiotherapy and surgery) is unclear as chemotherapy was repeated every 2 weeks for 4 to 6 cycles.
- The use of Student’s t-test to evaluate continuous variables in this manuscript in not appropriate as it is very unlikely that study groups of 15 participants would follow normal distribution with homogeneity of variances. All such variables should be re-evaluated with a non-parametric test.
- A treatment flow chart with appropriate timeline demonstrating intervals between individual therapeutic interventions should be provided to improve readability of both treatment regimens.
- It’s very hard to follow response to treatment. Four patients in the mFOLFOX-6 group achieved cCR, but only 17 were operated, leaving 1 without surgery and cCR. Table 4 shows details only for 17 operated patients. In the TEGAFOX group the authors reported one pCR. However, there were no cases of grade 4 TRG (i.e. complete regression)?
- Survival curves (figure 2) may be misinterpreted by readers. The median follow-up time was less than 3 years. However, both KM curves show 5-yr survival rates of 70-80% and the authors report 3-yr survival of more than 80%!
